# The Potential of *ANK1* to Predict Parkinson’s Disease

**DOI:** 10.3390/genes14010226

**Published:** 2023-01-15

**Authors:** Jinsong Xue, Fan Li, Peng Dai

**Affiliations:** School of Biology, Food and Environment, Hefei University, Hefei 230601, China

**Keywords:** *ANK1*, Parkinson, GEO database, biomarker

## Abstract

The main cause of Parkinson’s disease (PD) remains unknown and the pathologic changes in the brain limit rapid diagnosis. Herein, differentially expressed genes (DEGs) in the Gene Expression Omnibus (GEO) database (GSE8397 and GSE22491) were assessed using linear models for microarray analysis (limma). *Ankyrin 1* (*ANK1*) was the only common gene differentially down-regulated in lateral substantia nigra (LSN), medial substantia nigra (MSN) and blood. Additionally, DEGs between high *ANK1* and low *ANK1* in GSE99039 were picked out and then uploaded to the Database for Annotation, Visualization and Integrated Discovery (DAVID) for gene ontology (GO) functional annotation analysis. GO analysis displayed that these DEGs were mainly enriched in oxygen transport, myeloid cell development and gas transport (biological process (BP)); hemoglobin complex, haptoglobin–hemoglobin complex and cortical cytoskeleton (cellular component (CC)); and oxygen transporter activity, haptoglobin binding and oxygen binding (molecular function (MF)). Receiver operating characteristic (ROC) curve analysis showed *ANK1* had good diagnostic accuracy and increased the area under the curve (AUC) value when combined with other biomarkers. Consistently, intraperitoneal injection of 1-methyl-4-phenyl-1,2,3,6-tetrahydropy-ridi-ne (MPTP) in C57BL/6J mice reduced *ANK1* mRNA expression in both substantia nigra and blood compared to the control group. Thus, *ANK1* may serve as a candidate biomarker for PD diagnosis.

## 1. Introduction

Parkinson’s disease (PD) is a common neurodegenerative disease with complex etiology and pathogenesis. Studies have shown that α-synuclein accumulation, dopaminergic neuron loss, inflammation, oxidative stress and mitochondrial damage are involved in the process of PD [1,2,3,4,5]. However, the exact mechanisms of PD remain obscure and current treatments have limited effectiveness [6,7]. Moreover, PD is a debilitating disorder and characterized by severe motor (e.g., resting tremor, rigidity and hypokinesia tremor) and non-motor symptoms (e.g., cognitive deterioration, hallucination and sensory abnormalities) [8,9]. These symptoms progressively worsen over time and increase family burden [10].

It has been confirmed that PD is a progressive, disabling neurodegenerative disorder, with cases of either sporadic or familial origin. Sporadic PD is a multifactorial disease caused by both environmental and genetic factors. In addition, many genes have been identified in familial PD [11]. Numerous genetic loci associated with PD have been discovered. For example, 23 genes have been correlated with monogenic forms of PD in the Online Mendelian Inheritance in Man resource. Genome-wide association studies have helped researchers identify >90 risk loci associated with PD [12,13]. Studies from the Polish population have shown that four monogenic genes are involved in the PD pathological process, including parkin RBR E3 ubiquitin protein ligase (*PRKN*), phosphatase and tensin homolog-induced putative kinase 1 (*PINK1*), Leucine Rich Repeat Kinase 2 (*LRRK2*) and *α* synuclein (*SNCA*). Meanwhile, eight PD risk factor genes—namely, glucocerebrosidase (*GBA*), mitochondrial transcription factor A (*TFAM*), Nuclear factor erythroid 2-related factor 2 (*NFE2L2*), macrophage metalloelastase (*MMP12*), major histocompatibility complex, class II, DR α (*HLA-DRA*), Catechol-O-methyltransferase (*COMT*), monoamine oxidase B (*MAOB*) and dopamine β-hydroxylase (*DBH*)—were detected [12]. Therefore, genetic factors play a vital part in the development of PD and undoubtedly add to the complexity of this disease.

Timely diagnosis of PD may speed up identification, provide advance care planning, promote pharmacological interventions and delay debilitating syndromes [14]. Currently, PD diagnosis relies on clinical features, physical examination and disease biomarkers [7,15]. The diagnostic criteria for PD in the official International Parkinson and Movement Disorder Society (MDS) were proposed and based on the presence of Parkinsonism (i.e., bradykinesia with rest tremor, rigidity or both) [16]. Clinically established PD requires supportive criteria from at least two of the following: (1) clear beneficial response to dopaminergic therapy; (2) levodopa-induced dyskinesias; (3) rest tremor; (4) olfactory loss and cardiac sympathetic denervation [17]. Additionally, some structural imaging techniques have been employed to diagnose PD. For example, single-photon emission computed tomography of dopamine transporters (SPECT-DAT), a sensitive method with specific radio ligands, has been used to detect dopaminergic cell loss and track PD’s progression [18]. Magnetic resonance imaging (MRI) is a technique applied to appraise pathological changes of PD [19]. MRI is sensitive to neuromelanin, which decreases in substantia nigra (SN), and may be taken as an indicator of PD [20]. Accumulating evidence has shown that cerebrospinal fluid (CSF) and peripheral blood (PB) biomarkers increased the diagnostic performance. For instance, α-synuclein misfolding has a great relevance in the progression of PD and the measurement of its concentration is of clinical value for diagnosis [21]. Accordantly, *SNCA*, a gene that encodes human α-synuclein protein, was significantly down-regulated in the blood of PD patients [22]. Researchers have reported that *microRNAs*, including *miR-7-5p*, *miR-331-5p* and *miR-145-5p*, acted as effective biomarkers in PD diagnosis [23]. Previous meta-analysis of blood showed that hemoglobin- and iron-related genes, such as hemoglobin delta (*HBD*), α hemoglobin stabilizing protein (*AHSP*), ferrochelatase (*FECH*) and erythrocyte membrane protein (*EPB42*) were down-regulated in PD patients compared to controls [24]. Nonetheless, the symptoms of PD overlap with the features of other diseases, such as essential tremor, multiple system atrophy and progressive supranuclear palsy, making detection challenging [25]. Brain imaging techniques are costly and invasive [26]. α-synuclein and other candidates for PD markers are even less consistent [27]. Therefore, clinical diagnosis of PD is facing a bottleneck and searching for efficient biomarkers is an urgent and unmet need.

*Ankyrin1* (*ANK1*) is a large gene located on chromosome 8p11.21 and encodes the adapter protein family of Ankyrin-1 [28]. ANK1 was first identified to link integral membrane proteins to the underlying spectrin network [29]. ANK1 fulfills a vital role in maintaining the stability of the erythrocyte membrane. It has been reported that patients with hereditary spherocytosis may be caused by human *ANK1* mutation [30]. Mutation in *ANK1* can change red blood cells’ shape from a double-concave disc to a sphere and suppress its deformability [31]. Genome-wide association studies analysis showed the single nucleotide polymorphisms in *ANK1* (i.e., rs515071 and rs516946) have significant association with type 2 diabetes [32,33]. Additionally, several studies have demonstrated that *ANK1* showed differential methylation in Alzheimer’s disease (AD) and the cortical *ANK1* hypermethylation closely correlated with the presence of AD neuropathology. The up-regulated *ANK1* mRNA was found in microglia within the hippocampus of AD brains [34,35]. Recently, research has suggested that *ANK1* hypermethylation existed in the entorhinal cortex in AD, Huntington’s disease (HD) and PD, while no *ANK1* DNA methylation was found in the striatum in HD or the substantia nigra in PD [36].

In this study, using GSE8397 and GSE22491 as datasets, we discovered that *ANK1* was the common DEG in lateral substantia nigra (LSN), medial substantia nigra (MSN) and blood. Moreover, we investigated the PD-related gene changes and GO terms enriched in the DEGs (high *ANK1* vs. low *ANK1*). Further receiver operating characteristics (ROC) curve analysis and in vivo studies using 1-methyl-4-phenyl-1,2,3,6-tetrahydropyr-idine (MPTP)-induced mice confirmed *ANK1* may offer an efficient diagnosis of PD. Our findings shed light on the crucial role of *ANK1* in PD as well as provide a possible biomarker for predicting PD progression.

## 2. Materials and Methods

### 2.1. Microarray Data Information

Gene expression profiles of GSE 8397, GSE 22491, GSE 99039 and GSE 34287 were obtained from the Gene Expression Omnibus (GEO) database. GSE 8397 was sequenced on the platform of GPL 96 ([HG-U133A] Affymetrix Human Genome U133A Array), which included 15 PD MSN, 9 PD LSN, 8 control MSN and 7 control LSN samples. The platform for GSE 22491 was GPL 6480, Agilent-014850 Whole Human Genome Microarray 4 × 44 K G4112F (Probe Name version), which included 10 samples of PB from PD patients and 8 samples of PB from healthy controls. GSE 99039 (blood samples) from 233 healthy controls and 205 PD patients were downloaded from GPL 570, [HG-U133_Plus_2] Affymetrix Human Genome U133 Plus 2.0 Array. GSE 34287 (blood samples) was from GPL 7884, ExonHit Human Genome Wide SpliceArray 1.0 and included 19 PD samples and 12 normal samples.

### 2.2. Animals and Reagents

Male C57BL/6 mice (8 weeks) were purchased from Beijing Weitong Lihua Experimental Animal Technical Co. (Beijing, China). All mice were given free access to food and water and were kept on a 12-h light/dark cycle. The experimental procedures were performed according to ethical guidelines. 1-methyl-4-phenyl-1,2,3,6-tetrahydropyridine hydrochloride (MPTP-HCl) (Beyotime Institute of Biotechnology Co., Ltd., Cat#ST1020, Shanghai, China) was dissolved in saline. The following reagents were used: Trizol reagent (Invitrogen, Cat#15596026, Carlsbad, CA, USA), EasyScript^®^ One-Step gDNA Removal and cDNA Synthesis SuperMix kit (TransGen Biotech, Cat#AE311-02, Beijing, China), PerfectStart^®^ Green qPCR SuperMix kit (TransGen Biotech, Cat#AQ601-02, Beijing, China), 3-(4,5)-dimethylthiahiazo (-z-y1)-3,5-di-phenytetrazoliumromide (MTT) (Beyotime Institute of Biotechnology Co., Ltd., Cat#ST316, Shanghai, China), dimethyl sulfoxide (Biofroxx, Cat#196055, Guangzhou, Guangdong, China), Triton X-100 (BioFroxx, Cat#1139, Guangzhou, Guangdong, China), Tween 20 (Servicebio, Cat# GC204002, Wuhan, Hubei, China), rabbit anti-ANK1 antibody (Sigma, Cat#HPA056953, St Louis, MO, USA), rabbit anti-β-actin antibody (Servicebio, Cat#GB11001, Wuhan, Hubei, China), goat anti-rabbit antibody (Servicebio, Cat#GB23303, Wuhan, Hubei, China), FITC-labelled-goat anti-rabbit antibody (Biosharp, Cat#BL032A, Hefei, Anhui, China) and enhanced chemiluminescence substrate (Servicebio, Cat#G2020, Wuhan, Hubei, China). The animal procedures were approved by the Ethics Committee for Animal Experiments of Hefei University (No. 20RC49-202107).

### 2.3. DEGs Assessment

The downloaded dataset files were standardized by quantiles. DEGs between normal and PD samples were identified via the limma method. The screening criteria were |logFC (fold change)| > 1.5 and adjust *p* value < 0.05. The common DEGs were identified and visualized using heatmaps and Venn diagrams.

### 2.4. Protein–Protein Interaction Analysis

DEGs in GSE8397 and GSE22491 were transferred to the Search Tool for the Retrieval of Interacting Genes (STRING) database to construct a protein–protein interaction (PPI) network. Subsequently, Cytoscape software (v3.7.2, Cytoscape Consortium, San Diego, CA, USA) was applied to detect the correlations between DEGs. The Molecular Complex Detection (MCODE) plugin in Cytoscape was used to screen hub genes and extract functional modules. The settings were as follows: network scoring degree cutoff, 2; cluster finding, fluff; node density cutoff, 0.1; node score cutoff, 0.2; k-core, 2; max.depth, 100.

### 2.5. Genome Ontology (GO) Enrichment Analysis

GSE99039 datasets were chosen and the median value of *ANK1* was calculated. We divided patients into a high *ANK1* (h-*ANK1*) group and a low *ANK1* (l-*ANK1*) group according to the median value and DEGs were excavated. Gene ontology (GO) enrichment analysis is a commonly utilized technique for understanding the functional classification of all DEGs. The three GO categories, including biological process (BP), cellular component (CC) and molecular function (MF), were analyzed by the Database for Annotation, Visualization and Integrated Discovery (DAVID).

### 2.6. Receiver Operating Characteristics (ROC) Curve Analysis

GSE22491 and GSE34287 datasets containing blood samples were downloaded from the GEO database for drawing ROC curve (with sensitivity on ordinate and 1-specificity on abscissa). The ROC curve allows for the detection of the association between sensitivity and specificity. The validity and discriminant ability of genes for diagnosis were assessed using the area under the curve (AUC) value.

### 2.7. Construction of MPTP-Induced PD Model

Mice were divided into two groups: normal group (saline) and MPTP group (MPTP). Normal mice were given intraperitoneal saline injections. The PD construction was performed as previously described. Briefly, the mice received intraperitoneal injection of MPTP-HCl (dissolved in sterile normal saline, 30 mg/kg body weight) per day for 5 successive days. At 1 day after the last MPTP intraperitoneal injection, *ANK1* mRNA level was measured. The behavioral tests and protein expression assay were conducted 5 days after the last MPTP intraperitoneal injection.

### 2.8. Behavioral Tests

To test locomotor activity, open-field experiment was applied. Each mouse was placed individually in a box (50 cm length × 50 cm width × 50 cm height) with 16 equal squares. The spontaneous activity (i.e., crossings: the number of squares crossed; rearings: the counts of standing; groomings: cleaning frequency) were recorded for 5 min.

To measure motor coordination, the rotarod experiment was conducted to observe foot movement. Mice were placed on a rotating rod, which rotated along its longitudinal axis. We stopped the apparatus when the mice fell down or when the latency to fall reached 150 s. The length of time that the mice remained on the rotarod was recorded and used for measuring motor behaviors.

The spontaneous forelimb use was evaluated by cylinder test. Mice were placed individually in a glass cylinder for 10 min. We recorded the contact against the cylinder wall (i.e., rearing using the left forepaw, right forepaw or both paws together). The cylinder forelimb asymmetry ratio (CAR) was then calculated using the following formula: [(dominant forelimb touches + ½ number of both forelimbs)/(dominant forelimb touches + non-dominant forelimb touches + both) × 100] [37].

The suspension test was designed to measure grasping behavior. A 40 cm wire was fixed tightly in the platform and kept horizontally. Mice were put on the center of the wire and allowed to freely walk for a maximum of 1 min. The time spent by the mice hanging onto the wire was recorded and this index indicated muscle endurance [38].

### 2.9. Polymerase Chain Reaction (PCR) Analysis

Total RNA was extracted from mouse blood or SN using Trizol reagent, and cDNA was obtained with a reverse transcription kit according to the manufacturer’s protocol. The reaction mixture containing the template, forward primer, reverse primer and qPCR supermix was amplified using the LightCycler^®^ 96 Real-time PCR system (Roche). Relative mRNA expression was normalized to *β-Actin*. Sequences of primers were as follows: *ANK1* (forward: 5′-TGGAAGGAGCACAAGAGTCGT-3′; reverse: 5′-CAGAGCCAGCTT-CACTTTCTTG-3′), *β-Actin* (forward: 5′-GGCTGTATTCCCCTCCATCG-3′; reverse: 5′-CCAGTTGGTAACAATGCCATGT-3′).

### 2.10. Western Blot Analysis

Tissues were collected and homogenized in phosphate-buffered saline containing Triton X-100 (PBST). The protein samples were performed on sodium dodecyl sulfate-polyacrylamide gel electrophoresis and subsequently transferred to polyvinylidene fluoride membranes. The membranes were blocked for 2 h at room temperature using 5% skimmed milk. The membranes were then incubated with the primary antibody (anti-ANK1 and anti-β-actin) at 4 °C overnight and washed with PBS containing Tween 20 three times the next day. The blots were treated with secondary antibody for 2 h at room temperature. Finally, enhanced chemiluminescent substrate was used to visualize the bands. β-Actin served as loading control.

### 2.11. Immunofluorescence

Brain slices were blocked with goat serum, followed by incubation with the primary antibody (anti-ANK1). The slices were washed with PBS three times. The following step was incubated with FITC-labelled-goat anti-rabbit antibody for 30 min at 37 °C. After having been washed again three times with PBS, the specimens were sealed and photographed under fluorescence.

### 2.12. In Vitro Cell Viability Assay

Rat pheochromocytoma PC12 cells were cultured in Dulbecco’s modified Eagle’s medium (DMEM). The small interfering RNA (siRNA) against *ANK1* (sense: 5′-CACCCAAUGUCUCCAAUGUTT-3′, antisense: 5′-ACAUUGGAGACAUUGGG UGTT-3′) and negative control siRNA (sense: 5′-UUCUCCGAACGUGUCACGUTT-3′, antisense: 5′-ACGUGACACGUUCGGAGAATT-3′) were chosen to transfect PC12 cells. To measure cell viability, MTT solution (0.5%) was added into each well and incubated for 4 h at 37 °C. The purple formazan crystals that formed were dissolved in dimethyl sulfoxide. Cell viability was determined by measuring the absorbance at 490 nm.

### 2.13. Statistical Analysis

DEGs were generated using the limma method. ROC curve analysis was carried out and calculated through SPSS statistical software (version 23, IBM SPSS Statistics, Armonk, NY, USA). Data were expressed as mean ± SEM. Student’s t-test was applied to evaluate significant differences between two groups. *p* vlaue < 0.05 was considered significant.

## 3. Results

### 3.1. Identification of ANK1 as a Commonly Down-Regulated Gene

We collected nine PD LSN and seven control LSN samples from GSE 8397. We found 4 up-regulated genes and 41 down-regulated genes (Figure 1A). The GSE 8397 consisting of MSN (i.e., 15 PD MSN and 8 control MSN) was also analyzed and the generated heatmap showed 7 up-regulated genes and 47 down-regulated genes (Figure 1B). A microarray study of GSE 22491 on blood was chosen for further analysis. As shown in Appendix A, 114 genes were up-regulated and 56 genes were down-regulated. The exact data are shown in Appendix A. To excavate the overlapping genes, the Venn diagram was drawn to display the numbers of intersected gene sets. We observed that *ANK1* was the only DEG present (Figure 1C). For the comparison of LSN samples, *ANK1* was lowly expressed in PD, which was verified in MSN (Figure 1D–E). Consistently, *ANK1* in blood samples was also significantly decreased (Figure 1F). These results indicated that PD tissues may have low *ANK1* expression.

### 3.2. Construction of PD Associated Regulatory Network and Module Analysis

To better understand the interplay among the identified DEGs, STRING tool was used to elucidate protein–protein interaction networks. Four up-regulated genes and forty-one down-regulated genes from GSE 8397 LSN samples were submitted to STRING and a network was generated. The network was made up of three modules (13 nodes/30 edges, 7 nodes/8 edges and 5 nodes/5 edges, respectively) (Appendix A). A crucial module containing *ANK1* is shown in Figure 2A. In addition, we used the STRING database to analyze GSE 8397 MSN samples. The network contained 33 nodes. Two modules (11 nodes/24 edges and 15 nodes/21 edges, respectively) were extracted from the network (Figure 2B and Appendix A). Similarly, there were 97 nodes in the network of DEGs that were identified in our study (GSE 22491). A total of six modules were extracted (26 nodes/133 edges, 17 nodes/78 edges, 9 nodes/18 edges, 19 nodes/30 edges, 9 nodes/13 edges and 4 nodes/4 edges, respectively) and *ANK1* existed in two significant modules (Figure 2C and Appendix A). The GO enrichment analysis for these networks is presented in Appendix A. Taken together, these findings suggest that *ANK1* is associated with PD.

### 3.3. ANK1 Confers Significant Correlation with PD Progression

We selected GSE 99039, a peripheral whole blood microarray dataset with the biggest sample number, to further explore the association between *ANK1* and PD. Based on the median value, the GEO cohort was divided into two parts, namely, the high-*ANK1* (h-*ANK1*) group and the low-*ANK1* (l-*ANK1*) group. A volcano plot depicted the log2 fold change on the *x*-axis and the -log10 of adjusted *p*-values on the *y*-axis. The graph presented increased expression of 73 genes and decreased expression of 123 genes (Figure 3A). Consistently, Parkinson’s biomarkers—including membrane palmitoylated protein 1 (*MPP1*), solute carrier family 4, anion exchanger, member 1 (*SLC4A1*), erythrocyte membrane protein band 4.2 (*EPB42*), Selenium-binding protein 1 (*SELENBP1*), Glycophorin B (*GYPB*), α-Hemoglobin Stabilizing Protein (*AHSP*), hemoglobin-delta (*HBD*), α-synuclein (*SNCA*) and ferrochelatase (*FECH*)—were reduced in l-*ANK1* compared with that in h-*ANK1* (Figure 3B). To outline the functions of these DEGs, we performed a functional enrichment analysis (Table 1). GO BP analysis revealed that DEGs were mainly enriched in oxygen transport, myeloid cell development, gas transport and erythrocyte differentiation (Figure 3C). In the CC part, the DEGs were related to hemoglobin complex, haptoglobin–hemoglobin complex and cortical cytoskeleton (Figure 3D). Additionally, GO MF analysis demonstrated that DEGs participated in oxygen transporter activity, haptoglobin binding, oxygen binding and iron ion binding (Figure 3E). These mined BP, CC and MF terms have a close relationship with PD, which may explain the involvement between *ANK1* and PD.

### 3.4. ANK1 Shows a High Diagnosis Capacity in PD

To further evaluate *ANK1*’s potential application in the diagnosis of PD, we performed a ROC analysis to examine the sensitivity and specificity. Data from GSE22491 were analyzed and the results showed the AUC value to be 0.975, suggesting *ANK1* may be an effective biomarker for diagnosis (Figure 4A). In addition, we analyzed GSE34287 datasets and calculated the AUC value. ROC analysis for *ANK1* resulted in an AUC of 0.684. The AUC values for several PD-related genes, including zinc finger protein 134 (*ZNF134*), rho GTPase activating protein 26 (*ARHGAP26*), calcium voltage-gated channel subunit α1 D (*CACNA1D*), dopamine β-hydroxylase (*DBH*), epithelial membrane protein 1 (*EMP1*), interferon α-inducible protein 27 (IFI27), prenyl-binding protein phosphodiesterase 6D (*PDE6D*), proteoglycan 3, pro eosinophil major basic protein 2 (*PRG3*), solute carrier family 14 member 1 (*SLC14A1*), superoxide dismutase 2 (*SOD2*), transforming growth factor β 1 (*TGFB1*), thymocyte differentiation antigen 1 (*THY-1*), tripartite motif containing 24 (*TRIM24*), vesicle-associated membrane protein 4 (*VAMP4*), vesicle-associated membrane protein 8 (*VAMP8*) and zinc finger AN1-type containing 1 (*ZFAND1*), were 0.623, 0.643, 0.513, 0.579, 0.513, 0.746, 0.632, 0.667, 0.732, 0.544, 0.504, 0.553, 0.544, 0.649, 0.660 and 0.544, respectively. Furthermore, these genes with a combination of *ANK1* yielded an increased predictive accuracy of AUC values (*ZNF134*, 0.737, *ARHGAP26*, 0.759; *CACNA1D*, 0.706; *DBH*, 0.702; *EMP1*, 0.702; *IFI27*, 0.763; *PDE6D*, 0.737; *PRG3*, 0.702; *SLC14A1*, 0.763; *SOD2*, 0.711; *TGFB1*, 0.706; *THY-1*, 0.732; *TRIM24*, 0.702; *VAMP4*, 0.741; *VAMP8*, 0.724; *ZFAND1*, 0.728) (Figure 4B–Q). Collectively, these results confirm that *ANK1* has a good predictive power for PD’s diagnosis.

### 3.5. Validation of ANK1 in MPTP-Induced PD Model

To validate the predictive role for PD, we created a mouse model of PD by intraperitoneal injection of MPTP. A timeline of the experimental design is presented in Appendix A. Mice received a time course experiment and the rotarod test was applied at 1, 2, 3, 4 and 5 days after the last MPTP injection. As shown in Appendix A, at day 4 after the last MPTP injection, the latency to fall was significantly reduced compared with normal mice. Mice were subjected to behavioral tests, and blood and brain samples were collected for biological analysis. As indicated in Figure 5A, mice treated with MPTP exhibited a significant reduction in the number of rearings, crossings and groomings compared to the control mice. In addition, motor coordination was presented in Figure 5B, the latent period of PD mice to fall from the rotarod test was dramatically reduced compared to the control group. The calculated CAR from behavioral observation demonstrated that PD mice displayed higher forelimb asymmetry than normal mice (Figure 5C). Meanwhile, in the suspension test, the percentage of time spent on the wire was recorded, and MPTP administration considerably diminished this value (Figure 5D). To explore whether the MPTP-induced PD mice had a differentially expressed *ANK1* level, quantitative real-time PCR was operated and the result showed decreased *ANK1* mRNA expression in PD blood and SN (Figure 5E,F). The verification in vivo was in agreement with bioinformatic analysis in this study. Furthermore, MPTP treatment reduced ANK1 protein expression in blood and SN (Figure 5G). Immunofluorescence results showed that the green fluorescent signal intensity was decreased in the SN of MPTP group, indicating ANK1 expression may be down-regulated in PD (Figure 5H). In addition, in vitro assay indicated that knockdown of *ANK1* in PC12 cells can significantly reduce the cell viability compared to normal controls (Figure 5I). Considering that PD pathogenesis is characterized by neuron loss, the *ANK1* deficiency-induced PC12 cell injury further illuminates the potential correlation between ANK1 and PD.

## 4. Discussion

The discovery and validation of biomarkers are urgently needed for clinical performance improvement. Bioinformatics analysis on DEGs can provide potential molecules for diseases [39]. The lack of reliable and practical biomarkers is a major obstacle hindering accurate PD detection [40]. Given the potential severity of this condition, there is a growing requirement to engage research to identify novel assessments. In the present study, we initially obtained GSE datasets from the GEO database, derived from PD patients and healthy controls, and subsequently explored the DEGs and found that *ANK1* was commonly differentially expressed in all three tissues (i.e., LSN, MSN and blood). Thus, our research suggested that *ANK1* may serve as a biomarker for the diagnosis of PD and further analyses are needed to assess validity. PPI network analysis is an effective approach to identify hub genes with clinical merit [41]. Based on the above observation, a network analysis of DEGs was performed and key modules were extracted. We identified *ANK1* as a significant hub gene in the network of GSE8397, a dataset consisting of LSN and MSN. Similarly, in GSE 22491, six modules were gathered and we can see *ANK1* occupied two of these modules. Consistent with previous DEGs screening results, PPI network analysis showed that *ANK1* was in the key nodes and highly interacted with other genes.

To further illuminate the associations between *ANK1* and PD, we divided PD patients into two parts: an h-*ANK1* group and an l-*ANK1* group. We observed a significant down-regulation of genes, including *MPP1*, *SLC4A1*, *EPB42*, *SELENBP1*, *GYPB*, *AHSP*, *HBD*, *SNCA* and *FECH*. Among these genes, *SNCA* was identified as the first gene to be involved in PD [42]. It was proven that the relative *SNCA* transcript level in venous blood was considerably lower in PD patients compared with those in healthy subjects [22]. *MPP1* mRNA level has been reported to be decreased in two atypical parkinsonian disorders (multiple system atrophy (MSA) and progressive supranuclear palsy (PSP)) and PD compared with healthy controls. Thus, *MPP1* is a useful indicator to diagnose PD [43]. *SLC4A1* is responsible for anion exchanger 1 membrane protein production. According to the analysis of expression arrays, research has shown that the *SLC4A1* gene is reduced in PD blood versus controls [44,45]. Interestingly, we found a significant down-regulation of genes associated with hemoglobin and ion metabolism, including *HBD*, *AHSP*, *EPB42*, *SELENBP1*, *GYPB* and *FECH*. *HBD* encodes delta-chains of hemoglobin and a meta-analysis using blood samples was performed and reported that *HBD* was the second most highly ranked gene after *SNCA* [24]. Encoded by the *FECH* gene, ferrochelatase plays an essential role in maintaining iron metabolism [46]. Network-based meta-analysis identified *FECH* as the down-regulated gene in blood of PD patients [47]. *AHSP* encodes a molecular chaperone binding specifically to free α-globin and is necessary for hemoglobin assembly [48]. *EPB42*, a major component of the erythrocyte membrane skeletal network, engages in the maintenance of normal surface area in red blood cells and participates in iron homeostasis [49,50]. Human SELENBP1, a member of the selenium-binding protein family, is a highly conserved protein that covalently binds selenium and mediates the intracellular transport of selenium [51]. GYPB is glycoprotein of the human erythrocyte membrane and may represent changes in the synthesis of erythrocytes [52]. Our study observed that there was a significant decrease in the l-*ANK1* group’s blood *MPP1*, *SLC4A1*, *EPB42*, *SELENBP1*, *GYPB*, *AHSP*, *HBD*, *SNCA* and *FECH* genes compared with the h-*ANK1* group, which was consistent with the previous study that examined the differential blood gene expression between PD patients and healthy controls [24].

Hemoglobin is a protein heavily expressed in red blood cells. In mammals, the main function of hemoglobin is to transport oxygen from lungs to other tissues and interact with small functional molecules (e.g., carbon dioxide, carbon monoxide, and nitric oxide) [53]. As the most significant source of peripheral iron, hemoglobin may modulate iron homeostasis. Additionally, it has been proposed that PD incidence rose significantly as hemoglobin increased [54]. However, there exists evidence of a negative correlation between the hemoglobin level and PD. It has been proposed that PD patients with anemia or low hemoglobin may precede the motor symptoms by 20 or more years [55]. On the other hand, studies have shown iron homeostasis is dysregulated in PD patients [56,57]. The increased iron level has been found in the SN in PD patients in comparison to controls and high iron concentrations may be responsible for PD pathogenesis [58,59]. Intriguingly, increased serum iron may be associated with reduced PD risk, and therefore showing a protective effect of iron against PD [60,61]. Concurrently, most of the iron in the human body is present as heme iron and contained in the hemoglobin of erythrocytes. Its deficiency has been reported to link with anemia [62,63]. In this study, we divided the blood samples from GSE99039 into two parts (i.e., the h-*ANK1* group and the l-*ANK1* group) according to the median value of *ANK1*. We found genes associated with hemoglobin and iron homeostasis were down-regulated in the l-*ANK1* group. Furthermore, GO analysis has shown that these DEGs were enriched in GO terms related to oxygen transport, myeloid cell development, gas transport and erythrocyte differentiation (BP); hemoglobin complex, haptoglobin–hemoglobin complex and cortical cytoskeleton (CC); and oxygen transporter activity, haptoglobin binding, oxygen binding and iron ion binding (MF). These results demonstrate that *ANK1* may serve as a potential indicator for promoting PD diagnosis.

We further investigated the predictive performance of *ANK1* by drawing an ROC curve. In GSE22491, the AUC value was 0.975 and this result implies a high predictive ability of *ANK1*. In addition, GSE34287 was chosen and used to evaluate *ANK1*’s predictive effect. Although the single use of *ANK1* may not be effective enough in diagnosing PD, the combination of both genes resulted in improved diagnostic efficiency. For example, some genes such as *ZNF134*, *ARHGAP26*, *CACNA1D*, *DBH*, *EMP1*, *IFI27*, *PDE6D*, *PRG3*, *SLC14A1*, *SOD2*, *TGFB1*, *THY-1*, *TRIM24*, *VAMP4*, *VAMP8* and *ZFAND1* may improve PD’s diagnosis [47,64,65,66,67,68,69,70,71]. We have observed, in combination with *ANK1*, there was a significant increase in the AUC values of these biomarkers. As an integral membrane and adaptor protein, ANK1 modulates the attachment of membrane proteins (e.g., cell adhesion proteins and ion channels) and is pivotal for cell proliferation, mobility, activation and maintenance of specialized membrane domains [72]. Our results revealed that *ANK1* was correlated with PD’s potential factors, including the *SNCA* gene, hemoglobin related genes and ion metabolism. Thus, the application of *ANK1* for combined diagnosis may bring benefit in terms of increasing diagnostic accuracy.

Based on the previous results and analysis, we created an MPTP-induced mice PD model. MPTP is an analogue of the narcotic meperidine, which was found to induce parkinsonian-like syndrome [73,74]. The literature has substantiated that MPTP can easily cross the blood–brain barrier and finally be converted to 1-methyl-4-phenylpyridinium (MPP^+^). MPP^+^ destroys dopaminergic neurons via inhibiting mitochondrial complex Ⅰ and inducing oxidative stress [75,76,77]. In recent work, mice received an intraperitoneal injection of MPTP and showed reduced locomotor activity, impaired motor balance, elevated CAR and attenuated grasping strength. We were curious about whether different *ANK1* mRNA levels can be found between the control group and the MPTP-induced PD group. Consistent with the aforementioned data, the PCR results revealed that blood and SN *ANK1* mRNA expression was down-regulated in the PD group. Meanwhile, MPTP treatment reduced ANK1 protein level when compared to the normal group. Further studies have shown that ANK1 deficiency caused a decrease in PC12 cell viability, indicating the potential association between ANK1 and PD.

In summary, through performing bioinformatics analyses, including DEGs analysis, PPI network analysis and GO enrichment analysis, we identified that *ANK1* may serve as a biomarker for PD. Moreover, the AUC values we investigated and our in vivo studies suggest a good discriminative capacity of *ANK1* in predicting PD. Our research highlighted the potential clinical value of *ANK1* for PD diagnosis. Further studies considering the mechanisms undergirding the links between *ANK1* and PD are needed, which may shed light on PD prevention and treatment.

## Figures and Tables

**Figure 1 genes-14-00226-f001:**
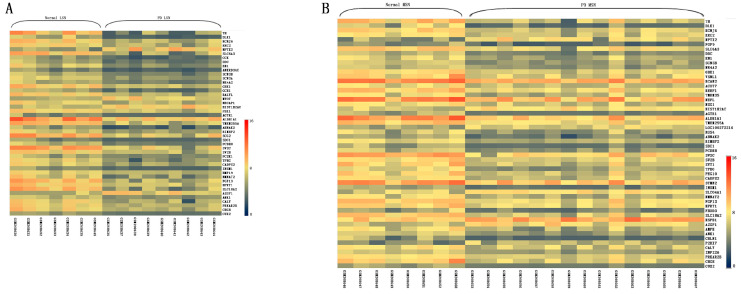
*Ankyrin 1* (*ANK1)* was identified as a down-regulated gene in Parkinson’s disease (PD) Gene Expression Omnibus (GEO) datasets. Differentially expressed genes (DEGs) expression heat maps in lateral substantia nigra (LSN) (**A**) and medial substantia nigra (MSN) (**B**). (**C**) Venn diagram of DEGs from GSE datasets. One gene was excavated as common DEG in three tissues, including LSN, MSN and blood. Differential expression of *ANK1* in LSN (**D**), MSN (**E**) and blood (**F**). Data are expressed as mean ± S.E.M. Two groups were analyzed by Student’s *t*-test. *** *p* < 0.01.

**Figure 2 genes-14-00226-f002:**
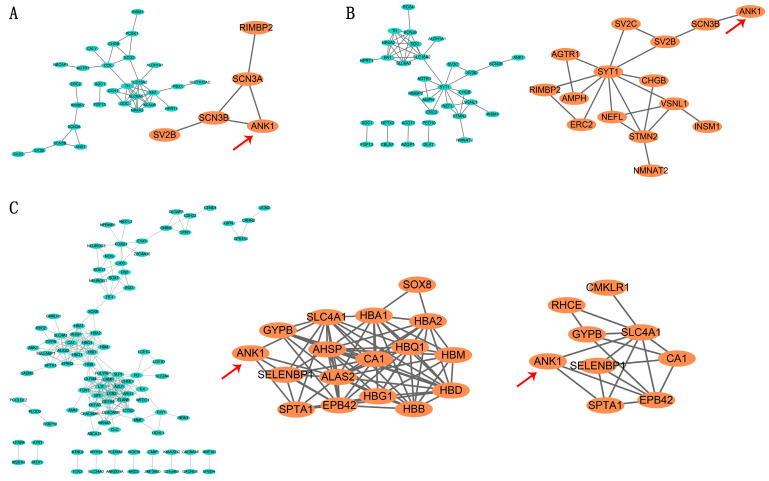
Protein–protein interaction (PPI) networks construction and module analysis. PPI networks construction and module analysis of GSE 8397 LSN DEGs (**A**), GSE 8397 MSN DEGs (**B**) and GSE 22491 blood DEGs (**C**). The nodes represent proteins and edges reflect protein-protein associations. The red arrows represent *ANK1*. The texts inside the elliptic nodes represent genes names. Light blue nodes and lines represent intact PPI network and the large orange nodes and lines indicate modules with *ANK1*.

**Figure 3 genes-14-00226-f003:**
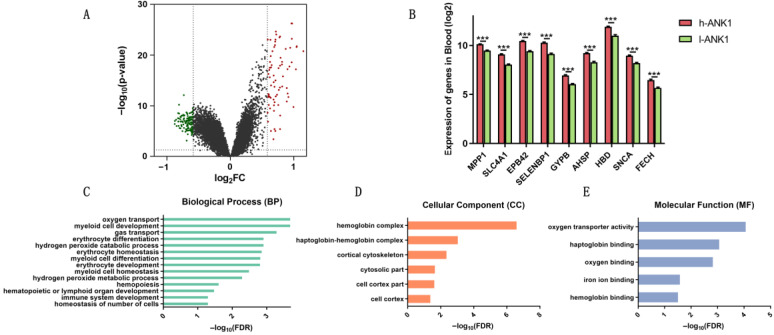
*ANK1* was associated with PD. (**A**) DEGs were extracted from GSE 99039 according to the median value of *ANK1*. Volcano plot of differential gene profiles between h-*ANK1* and l-*ANK1* group is shown. (**B**) PD biomarkers exhibited differential expression between h-*ANK1* and l-*ANK1* group. Based on *ANK1* expression level (high vs. low), the study analyzed the DEGs via function enrichment analysis. The enriched biological process (BP) (**C**), cellular component (CC) (**D**) and molecular function (MF) (**E**) terms are presented. Data are expressed as mean ± S.E.M. Two groups were analyzed by Student’s *t*-test. *** *p* < 0.01.

**Figure 4 genes-14-00226-f004:**
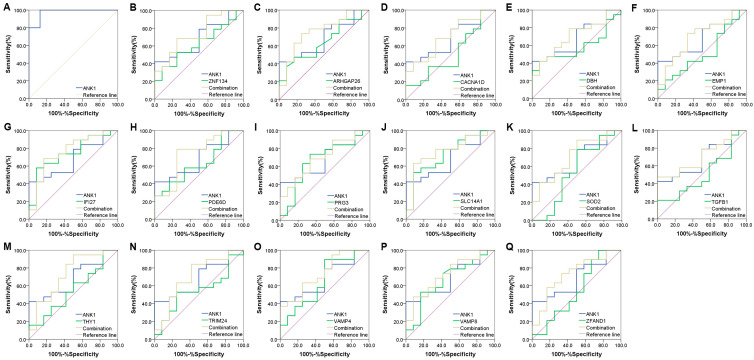
Diagnostic performance of *ANK1* in PD. (**A**) Receiver operating characteristics (ROC) analysis to evaluate the performance of *ANK1* as a diagnostic biomarker in GSE 22491. (**B**–**Q**) ROC curve to evaluate the diagnostic accuracy of *ANK1* and the combined diagnosis in GSE 34287.

**Figure 5 genes-14-00226-f005:**
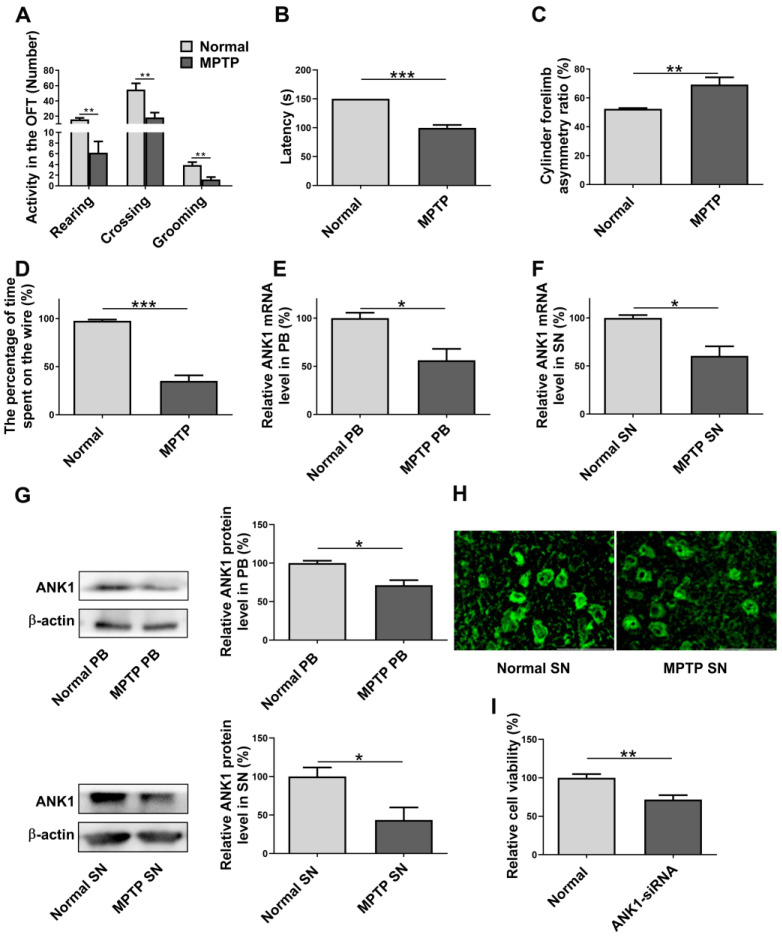
Validation of *ANK1* using in vivo and in vitro models. (**A**) The open-field experiment was achieved to test the mice’s spontaneous activity. (**B**) The rotarod test was carried out to estimate the motor balance. (**C**) The cylinder test was performed to appraise forelimb placement. (**D**) The suspension test was employed to measure muscle strength. (**E**) *ANK1* mRNA level in blood was measured by quantitative PCR. (**F**) *ANK1* mRNA level in SN was measured by quantitative PCR. (**G**) ANK1 protein expression was measured by Western blot. (**H**) Immunofluorescence of SN ANK1. (**I**) Cell viability analysis was performed by MTT analysis. Data are expressed as mean ± S.E.M. Two groups were analyzed using Student’s *t*-test. * *p* < 0.05, ** *p* < 0.01, *** *p* < 0.001.

**Table 1 genes-14-00226-t001:** Gene ontology analysis of differentially expressed genes associated with *ANK1* expression.

Category	Term	Gene Symbol	FDR
GOTERM_BP_FAT	GO:0015671_~_oxygen transport	*HBZ*, *HBM*, *HBD*, *HBQ1*, *BPGM*	0.000195516
GOTERM_BP_FAT	GO:0061515_~_myeloid cell development	*FAM210B*, *DMTN*, *EPB42*, *PIP4K2A*, *SLC4A1*, *BPGM*, *LTF*	0.000195516
GOTERM_BP_FAT	GO:0015669_~_gas transport	*HBZ*, *HBM*, *HBD*, *HBQ1*, *BPGM*	0.000489144
GOTERM_BP_FAT	GO:0030218_~_erythrocyte differentiation	*FAM210B*, *DMTN*, *EPB42*, *SLC4A1*, *AHSP*, *BPGM*, *KLF1*	0.001167109
GOTERM_BP_FAT	GO:0042744_~_hydrogen peroxide catabolic process	*HBZ*, *HBM*, *HBD*, *HBQ1*, *SNCA*	0.001167109
GOTERM_BP_FAT	GO:0034101_~_erythrocyte homeostasis	*FAM210B*, *DMTN*, *EPB42*, *SLC4A1*, *AHSP*, *BPGM*, *KLF1*	0.001324464
GOTERM_BP_FAT	GO:0030099_~_myeloid cell differentiation	*FAM210B*, *DMTN*, *EPB42*, *FAXDC2*, *PIP4K2A*, *SLC4A1*, *AHSP*, *BPGM*, *KLF1*, *LTF*	0.001445306
GOTERM_BP_FAT	GO:0048821_~_erythrocyte development	*FAM210B*, *DMTN*, *EPB42*, *SLC4A1*, *BPGM*	0.001476546
GOTERM_BP_FAT	GO:0002262_~_myeloid cell homeostasis	*FAM210B*, *DMTN*, *EPB42*, *SLC4A1*, *AHSP*, *BPGM*, *KLF1*	0.003097455
GOTERM_BP_FAT	GO:0042743_~_hydrogen peroxide metabolic process	*HBZ*, *HBM*, *HBD*, *HBQ1*, *SNCA*	0.004875849
GOTERM_BP_FAT	GO:0030097_~_hemopoiesis	*FAM210B*, *DMTN*, *EPB42*, *GLRX5*, *FAXDC2*, *PIP4K2A*, *SLC4A1*, *AHSP*, *BPGM*, *FBXO7*, *KLF1*, *LTF*	0.023463887
GOTERM_BP_FAT	GO:0048534_~_hematopoietic or lymphoid organ development	*FAM210B*, *DMTN*, *EPB42*, *GLRX5*, *FAXDC2*, *PIP4K2A*, *SLC4A1*, *AHSP*, *BPGM*, *FBXO7*, *KLF1*, *LTF*	0.031640479
GOTERM_BP_FAT	GO:0002520_~_immune system development	*FAM210B*, *DMTN*, *EPB42*, *GLRX5*, *FAXDC2*, *PIP4K2A*, *SLC4A1*, *AHSP*, *BPGM*, *FBXO7*, *KLF1*, *LTF*	0.048049424
GOTERM_BP_FAT	GO:0048872_~_homeostasis of number of cells	*FAM210B*, *DMTN*, *EPB42*, *SLC4A1*, *AHSP*, *BPGM*, *KLF1*	0.048049424
GOTERM_CC_FAT	GO:0005833_~_hemoglobin complex	*HBZ*, *HBM*, *AHSP*, *HBD*, *HBQ1*	2.25 × 10^−7^
GOTERM_CC_FAT	GO:0031838~haptoglobin-hemoglobin complex	*HBZ*, *HBM*, *HBD*, *HBQ1*	0.000819832
GOTERM_CC_FAT	GO:0030863_~_cortical cytoskeleton	*MPP1*, *TMOD1*, *DMTN*, *GYPC*, *EPB42*, *SLC4A1*	0.004003729
GOTERM_CC_FAT	GO:0044445_~_cytosolic part	*HBZ*, *HBM*, *GLRX5*, *AHSP*, *HBD*, *HBQ1*	0.020118051
GOTERM_CC_FAT	GO:0044448_~_cell cortex part	*MPP1*, *TMOD1*, *DMTN*, *GYPC*, *EPB42*, *SLC4A1*	0.021770885
GOTERM_CC_FAT	GO:0005938_~_cell cortex	*MPP1*, *TMOD1*, *DMTN*, *GYPC*, *EPB42*, *SLC4A1*, *SNCA*	0.037572121
GOTERM_MF_FAT	GO:0005344_~_oxygen transporter activity	*HBZ*, *HBM*, *HBD*, *HBQ1*	8.04107 × 10^−5^
GOTERM_MF_FAT	GO:0031720_~_haptoglobin binding	*HBZ*, *HBM*, *HBD*, *HBQ1*	0.000809657
GOTERM_MF_FAT	GO:0019825_~_oxygen binding	*HBZ*, *HBM*, *HBD*, *HBQ1*	0.00139728
GOTERM_MF_FAT	GO:0005506_~_iron ion binding	*HBZ*, *FECH*, *FAXDC2*, *HBQ1*, *LTF*, *SNCA*	0.024589133
GOTERM_MF_FAT	GO:0030492_~_hemoglobin binding	*SLC4A1*, *AHSP*, *HBD*	0.028801975

## Data Availability

Gene expression profiles were downloaded from the Gene Expression Omnibus (GEO) database (https://www.ncbi.nlm.nih.gov/geo/, accessed on 2021).

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
