# Peer review of "The Potential of ANK1 to Predict Parkinson’s Disease"

_genes, 2023, doi:10.3390/genes14010226_

Round 1

Reviewer 1 Report

Diagnosis of Parkinson's disease is challenging condition Xue et al. analyzed differentially expressed genes (DEGs) in Gene Expression Omnibus (GEO) database (GSE8397 and GSE22491) were assessed using linear models for microarray analysis. This is well designed study. Authors used proper methodology. I have only minor comments regarding the presentation of the results. Please remove red color of fonts. It would be useful to add short paragraph in introduction section about the kind of genetic causes of Parkinson's disease (Neurol Neurochir Pol 2021;55(3):241-252)

Reviewer 2 Report

The paper entitled “ANK1 Have the Potential to Predict Parkinson’s Disease” by Xue and coll, provides preliminary evidence of ANK1 mRNA level alteration in MPTP mouse model for Parkinson disease. Starting for a bioinformatic analysis, using gene expression profiles from the GEO database, Authors show that ANK1 has a “good predictive power for PD’s diagnosis”. Finally, by using MPTP mouse model Authors demonstrate a decrease in ANK1 mRNA level in both substantia nigra and peripheral blood. In other to achieve biological relevance this preliminary data, should be supported by functional experiments.

1)    There is no time course in the MPTP experiment: to suggest that ANK1 mRNA level is a diagnostic biomarker, it should be measured before the appearance of motor symptoms

2)    The decrease in mRNA level do not provide any evidence on protein levels and localization: a western blot and an immunofluorescence experiment is needed

3)     By using a cellular models Authors should demonstrate that alteration of ANK1 can recapitulate some of the PD cellular alteration(s)

Round 2

Reviewer 2 Report

Authors response to the major points raised, but the new data (WB, Immunofluorescence and PC12 experiment) should be included in the main text.
